# Convalescents’ Reports on COVID-19 Experience—A Qualitative Study

**DOI:** 10.3390/ijerph19106147

**Published:** 2022-05-18

**Authors:** Renata Bogusz, Luiza Nowakowska, Anita Majchrowska, Rafał Patryn, Jakub Pawlikowski, Anna Zagaja, Paweł Kiciński, Magdalena Pacyna, Elżbieta Puacz

**Affiliations:** 1Chair and Department of Humanities and Social Medicine, Medical University of Lublin, 7 Chodźki St., 20-093 Lublin, Poland; renata.bogusz@umlub.pl (R.B.); luiza.nowakowska@umlub.pl (L.N.); rafal.patryn@umlub.pl (R.P.); jakub.pawlikowski@umlub.pl (J.P.); anna.zagaja@umlub.pl (A.Z.); 2Department of Experimental Hematooncoloogy, Medical University of Lublin, 1 Chodźki St., 20-093 Lublin, Poland; pawel.kicinski@umlub.pl; 3Regional Center of Transfusion Medicine, 8 Żołnierzy Niepodległej St., 20-078 Lublin, Poland; magdalenapac@wp.pl; 4Department of Laboratory Diagnostics of SPZOZ, 4 M. Sobieskiego St., 22-300 Krasnystaw, Poland; elapu@poczta.onet.pl

**Keywords:** COVID-19, convalescents, patient’s role, experience of the disease

## Abstract

Background: The dynamic character of the COVID-19 pandemic and its social consequences caused several medical and societal issues and dilemmas. The aim of our qualitative research was to capture and analyze attitudes and beliefs of convalescents who experienced mild symptoms of COVID-19 in the first wave of the pandemic and decided to donate their plasma for therapeutic purposes. Material and Methods: The article presents results of qualitative research conducted on the basis of grounded theory (GT) methodology. Empirical material includes 10 in-depth interviews conducted with respondents who had mild or asymptomatic disease and, after recovery, voluntarily donated their plasma to the Regional Centre for Blood Donation and Blood Treatment (RCKiK). Data were collected in May and June 2020 in Poland. Qualitative analysis was focused on the experience of convalescents who entered the social role of a sick person in individual, social, and organizational dimensions. Results: The social role of the patient in the narratives of convalescents was related to three stages: (1) initiation to the role, (2) staying in the COVID-19 patient role, and (3) leaving the role. Research results enabled the distinction of three basic descriptive categories (“ontological uncertainty”, “the global and individual dimension”, and “being sick in the disease-infected environment”), which became epistemological framework for a detailed description of the roles played by an individual COVID-19 patient during the pandemic. Conclusions: The disease, despite its mild course, generated a number of non-medical issues, and the entire process of being ill was burdened with institutional and emotional struggles. The experience of mild COVID-19 is significantly modified by disease institutionalization. These results may contribute to a better understanding of the psychosocial dimension of COVID-19 and convalescents’ motivations for plasma donation.

## 1. Introduction

Within the past two years COVID-19 has turned into a global pandemic. Its scope, dynamic nature, and health consequences for individuals, even in case of a mild form of the disease, has generated a number of both medical and non-medical problems.

The health consequences of recovering from COVID-19 are extensively studied, however, it is also worth focusing on the psychosocial consequences of experiencing the pandemic on both an individual and macro-social scale. As indicated by several studies, psychological problems are a common experience of many COVID-19 convalescences [1,2,3] Testing positive for COVID-19 entailed fear, uncertainty, and anxiety about one’s own health and the future [4] even in the absence of severe symptoms. In addition, patients, especially from the first period of the pandemic, faced social stigma and stigmatization of being infected with the SARS-CoV-2 virus [5]. Sometimes, they experienced a sense of social rejection, even from medical institution staff [6]. Loss of sense of control, shame, and a sense of social isolation are the experiences reported by patients of the so-called first wave of COVID-19 [7]. Despite negative experiences, some of the convalescents decided to donate their plasma for the treatment of others. Therefore, it seems reasonable to analyze the experience of the disease from the psychosocial perspective in this particular group of patients who transformed their experience from negative to positive.

Due to all this, experience of COVID-19 can be perceived in socio-medical terms. The reference framework for this perspective may be the so-called “humanistic sociology”. Its representatives in the sociology of medicine include authors such as J. Roth [8] and E. Freidson [9]. However, the 1975 publication by AL Strauss and BG Glaser titled “Chronic Illness and the Quality of Life” [10] was of particular importance in the creation of this article. Its authors analyzed the subjective experiences and meanings that respondents attributed to chronic illness. Since that publication, aside from the structural–functionalist interpretation of health problems represented, e.g., by T. Parsons [11], the context of everyday life and valuing of the perspective of an ordinary person (interactionist and phenomenological paradigm) was also taken into account. This approach enables the recognition of sequences of behaviors spread over time and the identification of mechanisms of the disease’s impact on all dimensions of the individual’s functioning [10,11,12].

The research was conducted on the basis of Grounded Theory (GT), which is currently one of the most frequently applied qualitative research methodologies in the area of health and illness. This methodology forms a particularly valuable procedure that allows us to capture and comprehend the experience of individuals coming into contact with the healthcare system, as well as to explain key processes “occurring” in healthcare from the perspective of health services users [13]. The basic shape of GT was laid out by B. Glaser and A. Strauss in their 1967 work titled “The discovery of grounded theory. Strategies for qualitative research” [14]. This methodological orientation is aimed at developing categories based on collected data and therefore rejects starting form preconceived hypotheses. Hence, gradually, with each stage of the research, the final theory emerges. Within the framework of grounded theory, the applied procedure involves making notes in order to develop categories, indicating their properties and their relation to other categories, as well as adding samples to refine the theory. Our present analysis refers to the constructivist variation of GT, as proposed by K. Charmaz, and is focused on creating adequate categories, rather than on applying the available theoretical concepts [15]. As opposed to the theory of B. Glaser, A. Strauss assumes that theories created through the process of analysis do not uncover objective reality, but are a “product” created in the research process, influenced by both the researcher and the respondent.

The article’s objective was to capture and analyze attitudes and beliefs of convalescents who experienced COVID-19 symptoms in the first stage of the pandemic and then contacted the Regional Centre for Blood Donation and Blood Treatment to donate their plasma for therapeutic purposes. We mainly focused on the experience of convalescents related to entering the social role of a sick person in individual, social, and organizational dimensions. The analysis considered the classic concept of the patients’ role, initially introduced by T. Parson [11].

## 2. Material and Method

### 2.1. Study Procedure

The study procedure was initiated after obtaining a positive opinion of the bioethics committee (KE-0254/155/2020). The research was carried out among COVID-19 convalescents who volunteered to donate their plasma at the RCKiK (Regional Centre for Blood Donation and Blood Treatment) in Lublin. The type of sampling was purposive. Research participants were recruited among people who, by giving their written consent to donate biological material, also agreed to participate in social research, sharing their telephone numbers with the research team. Interviews were arranged individually with each of the respondents. The inclusion criteria were being COVID-19 convalescent as documented by a PCR (Polymerase Chain Reaction) genetic test (positive result at diagnosis, double negative results as evidence of recovery), donating plasma for therapeutic purposes, and giving the consent to participate in our research.

In-depth telephone interviews were carried out on respondent-selected days. Each respondent consented to the recording of interviews and use of the recorded material for scientific purposes. In addition, selected data on convalescents were also collected, protecting their anonymity. The first interview was carried out on 7 May, and the last on 13 June 2020. A total of 6 h 39 min of conversations were recorded, totaling 119 pages of verbatim transcription.

Collected data were subjected to comparative analysis on an ongoing basis. After reaching the saturation of the emerging categories, we decided to stop interviewing and include 10 reports of COVID-19 experience in the research material. Theoretical saturation is an important methodological procedure that determines completion of data collection; the researcher abandons adding further comparison groups when the emerged categories are saturated, which signifies that no additional new data appear and repetition of examples occurs [14].

Preliminary coding succeeded transcription, which was then followed by concentrated coding that allowed for the elaboration of categories and subcategories and identification of the correlations between them. The results are presented in the form of a graphic diagram and were supplemented with quotes from respondents’ statements. No computer programs were used to compile the data.

### 2.2. Participants

A total of 10 people (3 women and 7 men), all with an official COVID-19 convalescents status, participated in our study. The age of the respondents ranged from 21 to 59 (mean 35.9), with only one person over 50 years of age. Two of the respondents had basic vocational education, four had secondary, and five had higher education. Of these, 7 lived in blocks of flats in an urban area (including 5 living in a large city) and 3 in rural areas—in detached houses. The majority of the respondents (N = 8) lived with their family members (parents and siblings; spouse and children; only a child or a spouse, children, and parents/in-laws). One person shared an apartment with a roommate and one with a partner. All of the respondents were professionally active (two were also studying). At the time of the interview, only one person remained on sick leave, the rest had already returned to work. The respondents assessed their financial situation as good. The first positive COVID-19 test result in the research group was recorded in mid-March (14 March 2020), the last one in mid-April (15 April 2020). Obtaining a second negative test result was interpreted as confirmation of recovery (in the initial stage of the pandemic, it was necessary to obtain two negative test results for the person to be officially declared healthy). Duration of illness, counted from the moment of obtaining a positive result to the second negative one, was 13 to 41 days.

## 3. Results

The result of the analysis comprise a description of the first-wave COVID-19 convalescents’ experiences. It has been presented in a dynamic manner as a specific process of functioning in the role of a patient. Respective stages consist of a number of detailed categories that explain how the process of being ill was experienced and the status of convalescent attained.

### 3.1. Stage 1: Initiation to the Role of a Patient

#### 3.1.1. Becoming Aware of the Infection

The initial moment for the initiation to the role of the patient was the moment of infection. As it was impossible to precisely identify the time and place of infection, respondents were not always aware of their health situation. From this point of view, it is difficult to consider infection as a perceived and experienced moment in the disease’s course; however; we decided to include the category of infection as the starting point for the initiation stage, as it was connected with ex-post reflections. Becoming aware of the infection, i.e., by observing symptoms, is certainly more evident as a beginning of the initiation stage.

Becoming aware of the infection was related to recognizing the symptoms of the disease in oneself and linking them to COVID-19. Here, discrepancy was observed; some of the respondents associated the experienced symptoms with this disease with high probability from the very beginning, especially when they knew that it occurred in their immediate environment. However, the observation of symptoms was not always so clear and the respondents did not always immediately associate their health condition with COVID-19. In this scenario, the respondents became aware that the experienced symptoms could be caused by SARS-CoV-2 with time.

#### 3.1.2. Actions Implemented

Respondents, taking into account the possibility of being infected, took specific actions. These actions were therefore a reaction to the occurrence of disturbing symptoms, or to obtaining information that someone in their environment was quarantined or had a positive test result. In the analyzed statements, a two-fold path of taking actions was observed. Action was undertaken either by the patient (contact with a GP or the State Sanitary Inspection, self-isolation) or the party activating or coordinating such tasks, i.e., workplace superiors or the State Sanitary Inspection (SSI) (COVID test or quarantine referral). Importantly, when the action was initiated by the respondents themselves, various types of deficiencies and system shortcomings, and even conflicts related to inclusion in the health care system, were evident:

*In the first week there was fever (…) muscle pain. (…) we were told that the fever was too low, we do not have typical symptoms, so it was definitely not that (…) later, I lost my sense of smell and taste completely (…) I called the SSI* [State Sanitary Inspection-ed], *where a lady told me that it could be sinusitis. So we still haven’t been tested. (…) And after 2 weeks my mother started to have fever, cough. And only then Inspection decided to test us and it turned out that the results were positive*.[C 9, F]

#### 3.1.3. Emotional Reactions

The first spectrum of emotions experienced by respondents was strictly related to allowing the thought of the possibility that one might be infected. These fears intensified due to the sense of responsibility for relatives or colleagues:


*(…) I was stressed (…) I have a disabled mother. I was less worried about myself, than about my mother (…) more about my parents because they are elderly (…).*
[C 2, F]

Another moment capturing how the respondents experienced this early stage of functioning in the disease is their waiting for the test result. This was a particularly critical time. On one hand, it was emotionally difficult because the diagnosis could be confirmed. On the other hand, emotions were paradoxically softened by the awareness of the high probability of being sick, which meant “getting used to” the role of being diagnosed with COVID-19. In this situation, the awareness of being ill, in a sense, occurred even before medical diagnosis confirmed it. It seems that this ex ante self-diagnosis resulted in an alleviation of the tension associated with waiting for the results of laboratory tests:


*(…) I must admit, I was a bit scared. However, once I lost my taste and smell, it had come to me a bit, I calmed down.*
[C 6, M]

### 3.2. Stage 2: Staying in the COVID-19 Patient Role

#### 3.2.1. Positive Test Result

Getting a positive test result initiated a number of activities and developed new areas of dilemmas related to being ill. This is illustrated by one of the respondents:

*They took my swab, and on Thurs**day morning they called me saying I’m positive (…) and then the snowball effect continued*.[C 1, M]

We identified two forms of responses to a positive test result with different degrees of surprise. People from the so-called “contact” groups, that is, having contact with a previously confirmed case, had grounds for suspecting that their result may be positive, anticipating the role of the patient:


*(…) the information that I am positive (…) did not surprise me in any way.*
[C 3, M]

However, there were also reports of situations where, with few or no perceived symptoms, a positive test result came as a complete surprise:


*I was rather calm (…) I was sure that the result would be negative and that I would eventually go home.*
[C 7, M]

One of the narratives identified a special situation of complete “disharmony” of the role of COVID-19 patient, when obtaining information about a positive test result occurred after the disease process was completed, and its symptoms resolved. This situation relieved the respondents form an emotional baggage associated with being diagnosed with COVID-19:


*Somehow it happened that we were not aware that we have COVID. We went through it like through a common cold; the seasonal flu, and that was it. There were no pigs flying, or anything like that.*
[C 7, M]

#### 3.2.2. Confrontation with Institutions

Due to the fact that COVID-19 grew to the scale of a global pandemic, we can talk about the institutionalization of the disease process. At the macrosocial level, procedures for managing the disease and contacting individual institutions involved in the process of diagnosis, treatment, and recovery were all established. Confrontation with medical institutions, and various associated difficulties, came to the fore in the analyzed narratives at various stages of the illness process. At the stage of being ill, it was the struggle with institutions, and not with the symptoms of the disease, that became the dominant experience of the process of being ill/being diagnosed as positive. Such a situation was certainly due to the fact that the respondents were generally in good health and experienced relatively mild symptoms of the disease.

The State Sanitary Inspection (SSI) played the dominant role in the process of standardizing COVID-19-related behaviors, deciding to quarantine, test, isolate, or refer to hospitals those who were suspected of being infected or were infected with the SARS-CoV2 virus. The assessment of contacts with PIS and the perception of its functioning is therefore extremely important for describing experiences related to being ill. The narrative reveals a threefold character of this assessment: positive, negative, and negative coupled with understanding/justification. The positive assessment was conditioned primarily by satisfaction with the quality of information obtained from SSI:


*(…) my phone calls to the PIS (…) no matter what I was asking about, I never met with refusal, snapping back, or lack of assistance, there was always someone trying to help me somehow.*
[C 7, M]

In narratives, however, the assessments leaning towards the negative was more often identified. Negative assessments seemed to have a similar background, related to the information chaos or even complete lack of necessary information, the deficit of which significantly hindered functioning as a patient and undertaking advised actions. The respondents’ statements also reveal an unprofessional approach of the employees of SSI, their lack of empathy and commitment to providing competent information, or simply the lack of communication skills that would allow the communication to be adapted to the recipients’ needs.


*We were scared; the lady from SSI was shouting at me, asking how is it possible that my son got infected and other children from his class did not.*
[C 9, F]

The new category of evaluation that appeared in the respondents’ statements was the “negative-understanding/justifying” evaluation. This was expressed as a verbalized criticism of the institution’s functioning, its unprofessional approach or mistakes made by its employees, while pointing towards the causes of this situation and expressing some justification for it:


*(…) the whole situation was new for everyone, as well as for them [SSI-ed] and for all services, so it was an amazing mess. (…) they were overwhelmed with calls, they were unable to cope with them, they sometimes didn’t simply know how to help and what to do next.*
[C 1, M]

An important element in formulating a “negative-understanding/justifying” assessment was the passage of time and the reflection on the situation from the convalescent’s point of view:


*(…) then you have to understand that during that time (…) there were a lot of infected people and a lot of people in quarantine, so… (…) because it looked a bit different then, from a different perspective. Now, when I left there and I am a healthy person, I begin to think differently, begin to understand what was happening there, what we expected, and that it maybe was a bit too much.*
[C 5, M]

Medical institutions, such as hospitals and primary healthcare entities, also played an important role in the environment of COVID-19 patients. The course of the disease in most of the convalescents was mild, with no life-threatening symptoms or serious fear of loss of health, hence the context of relations with these institutions was not marked with strong emotions. Respondent’s assessment of medical institutions was rather positive, and it took into account the attitudes of doctors and nurses, and to a lesser extent the procedures, treatment methods, or organization of medical assistance. The relational element came to the fore in the accounts of convalescents, their sense of securing their emotional needs, and the concern for health and well-being expressed by medical personnel was obscured by the perceived dysfunctions, such as the lack of standardized procedures in the hospital setting:


*(…) the doctor (…) always called in the morning, asked about the temperature, blood pressure, heart rate, and then informed me what medications I would be administered today and how the treatment would be proceeding.*
[C 3, M]

#### 3.2.3. Social Isolation

Isolation was a common experience for all COVID-19 patients. Statutory law required isolation, although in practice its scope varied, depending on the individual situation of the patient and his/her family. The experiences of people staying in quarantine at home with their families were extremely different when compared to the experiences of those staying alone in the so-called isolation wards, and led to different adaptation and coping strategies. Respondents’ statements demonstrate that it was the time of staying in isolation from loved ones that was one of the most difficult experiences that patients encountered. Staying in the isolation ward (longest stay 26 days), in a single, usually small room, with no personal contact with anyone, except for medical personnel dressed in coveralls and masks, required looking for remedies to tame this isolation. What proved as the most effective strategy was the search for daily routine; the attempt to keep a normal daily schedule in this extraordinary situation.


*(…) I organized the day for myself so that I would get up at eight or seven (…) after breakfast, I read a book, then did some physical exercise (…) then read a book again, had lunch, and after lunch, maybe watched a movie (…) I also tried to stick to my own schedule, so to speak. (…) And it was easier if you planned your whole day there and stuck to that plan.*
[C 5, M]

#### 3.2.4. Coping Strategies

Despite the relatively mild course of the disease, the duration of illness was not devoid of emotions. Emotional peace, although it rarely appeared in the respondents’ narratives, occurred when the symptoms were not evident or when they were “unaware” of the disease because confirmation of a positive test result was obtained only after “getting sick” and after symptoms disappeared:


*(…) apart from the nervousness connected with the need to spend another two weeks in quarantine (…) it’s basically nothing. (…) We were all already healthy (…) when we had positive results, we were at the end of the disease.*
[C 7, M]

Most often, however, uncertainty, fear for one’s own health and that of loved ones, exacerbated by information chaos, the deficit of knowledge about various aspects of being ill, and the feeling of unpredictability of the whole situation, disturbed the sense of both health and social security, triggering negative emotional reactions:


*I had darkness before my eyes, back then no one knew much about this disease. I didn’t know what would happen to me; nobody knew, (…) how it would happen, what would happen with me, how many people I infected (…) So many unpleasant emotions.*
[C 2, F]

Although being a COVID-19 patient, as demonstrated by the reports of convalescents, was a completely new experience, both on an individual and macrosocial scale, the strategies for coping with the emotions accompanying the disease were the same as in other crisis situations. The most frequently applied strategy was to seek support from primary systems, i.e., family, friends, and close acquaintances. Contact with relatives and conversation, sometimes only by phone during isolation, allowed the possibility to “tame the stress” caused by the situation. We also noted a situation in which the disease became an opportunity to strengthen bonds and improve family relationships.

An interesting category that emerged from conversations with convalescents was a form of oversaturation with support from relatives, which caused a feeling of being overloaded by the role of patient. Situations such as this were caused by excessive—in the opinion of the respondents—interest in their well-being or health through an overwhelming number of questions and phone calls from their family members and friends:


*I had a lot of such conversations, and friends from work were calling every now and then, and I had to repeat the same to everyone. (…) it is not that a person is malicious or something, but it was the same thing over and over again. “Well, I’m fine, all right. I don’t know what the test results are. No. I don’t know when I will be able to leave.” And so on.*
[C 8, M]

The narrative also reveals a special way for dealing with stress, anxiety, and negative emotions. It is the pandemic information blockade, avoiding news about the virus and blocking any external information (mainly media) regarding COVID-19: 


*I tried not to read any messages or look at any information about it because it made me even more stressed.*
[C 2, F]

### 3.3. Stage 3: Leaving the Role of Patient

#### 3.3.1. Coming Back to Normality

Interviews were conducted shortly after the respondents’ illness, and although in general this phase of convalescent experiences was characterized by the slow stabilization of their health and return to their former roles, some still experienced negative consequences of the infection. Therefore, leaving the role of a patient was not definitive for everyone; sometimes, on the contrary, it was extended in time.

Sometimes, convalescents perceived the moment that formally determined the end their isolation as the time of real recovery:


*I was released home from the isolation and recognized as a person who was already healthy, and I also felt as if nothing had happened. I didn’t experience any symptoms myself, everything was fine.*
[C 4, M]

In other cases, despite the second negative test result, and even during the interviews, the respondents believed that symptoms typical of COVID-19 (e.g., dyspnea, impaired sense of smell and taste) persisted, that previous health issues (e.g., allergy) were exacerbated, or they experienced reactions to the strong stress they went through (e.g., difficulties falling asleep). Moreover, some convalescents, concerned about their own health, chose to undergo additional consultations (e.g., ear, nose, and throat examinations, blood tests, and X-rays of the lungs) or used dietary supplements as part of self-treatment to strengthen their body’s immunity. Furthermore, a few weeks after leaving the isolation wards, uniformed services employees who suffered asymptomatic disease were scheduled for comprehensive checkups in the hospital:


*Theoretically, it was all over for me. Then, even after the negative results, I felt, deep in my chest, that it was much harder to breathe, but that was gone too. On the other hand… my allergy exacerbated.*
[C 10, F]

Despite obtaining the convalescent status, respondents still feared for their own and their relatives’ health. They were not sure if they constituted a threat to someone or whether they could be re-infected. It was obvious to everyone that by contacting someone or something infected, contribution would be made to the spread of the virus. The experience and the circumstances of being ill, as perceived from an individual and social perspective, prompted the respondents to be highly cautious—even more so as it was still a time during which numerous places and institutions were subjected to the lockdown:


*In the beginning, I wasn’t seeing anyone, not a person. I didn’t even see my own boyfriend for a month… I didn’t leave the house. Only after a month did I go out for some short shopping. But then I was so stressed out, because people were still afraid then.*
[C 2, F]

#### 3.3.2. Social Relations

A wide variety of examples related to changes in interpersonal relationships after the disease appeared in the narratives of convalescents. Depending on the environment mentioned by the respondents (family, neighborhood, or professional), positive, negative, or no changes were observed.

In the respondents’ opinion, the prevailing view was that in the immediate family, their disease did not alter previously existing relationships. Sometimes, however, convalescents were surprised by the extent of help and support they obtained from their loved ones.

Respondents inhabiting city areas did not notice differences in neighbor relationships even despite their supposition that the neighbors knew about the infection. Residents of smaller communities were sure that information about their being afflicted spread quickly, and the extent of their sickness was even exaggerated by their neighbors and friends. As a consequence, convalescents regretted to observe distancing that resulted from fear of them, or even of their families:


*Here people are afraid, they are still really afraid.*
[C 9, F]

Relations at work, following illness, depended on the number of infected people in the workplace. In many situations, being ill together brought people closer and helped them face the difficult time of isolation:


*(…) our relations are good, very good. I think they’re even better now between those who got infected. We supported each other, we called each other, we talked about the symptoms.*
[C 1, M]

Single cases in the workplace became a reason for inquiries (e.g., when someone previously did not believe in the existence of the virus), but also compelled co-workers and the convalescents themselves to keep their distance:


*(…) at work from a distance, this is how we talked to each other—from a distance, because they knew that I was sick. Well, that’s how they try to avoid me.*
[C 8, M]

#### 3.3.3. Decision to Donate Plasma

The last stage characterizing the role of the patient in the narratives of convalescents was related to the donation of plasma for research or therapeutic purposes. Some of the respondents initially learned about this possibility from media broadcasts. On the other hand, some of them were informed and encouraged to take part in the procedure by SSI, medical professionals or colleagues. Sometimes there were further phone calls from medical institutions in order to clarify the terms. In other situations, convalescents contacted the RCKiK on their own initiative.

According to the respondents, expressing their consent to donate plasma was something natural and obvious for them (some of them had a history of blood donations). Possible concerns arose, however, in the case of breastfeeding or older age and diagnosed hypertension. The unequivocal motivation for everyone was the willingness to help others: those who were more ill, patients in life-threatening situations, but also their own parents, who may become ill in the future. The respondents also paid attention to the innovativeness of the procedure, and the fact that, to date, only some recovering patients could donate plasma:


*What made me do this? Just a willing to help. Well, if someone who is not so lucky and this disease affects him in a different way than me, i.e., has some more serious symptoms, something happens that his health, and perhaps even life, is at risk, and my time devoted to donating plasma can help with this and save someone, so it doesn’t cost me a lot of effort or an enormous amount of time, so why shouldn’t I agree?*
[C 4, M]

## 4. Discussion

The presented analysis describes the dynamic stages and specific dimensions that we identified in experiencing the role of the patient, based on the accounts of COVID-19 convalescents (Figure 1). Our respondents had mild or asymptomatic experiences with the disease, which, combined with individual motivations, allowed them to donate plasma. This in turn contributed to the development of innovative research into COVID-19 treatments [16,17,18,19,20].

We decided that it was worthwhile to take a closer look at this group for two reasons. Firstly, our decision was influenced by the availability of respondents recruited from RCKiK in the city that was the first in Poland to initiate plasma treatment of COVID-19 patients. Secondly, we decided that research should not only apply to patients with a severe course of disease, as this was already undertaken by numerous authors [21,22,23,24,25]. The latter argument is additionally supported by the fact that, as demonstrated by global statistics in the first period of pandemic in 2020, the overall hospitalization rate for COVID-19 patients was not high [25]. According to the Ministry of Health, in Poland, it was 9.78 per 100 thousand inhabitants, with Slovakia recording the lowest in Europe (4.27), the United Kingdom the highest in Europe (120.5) [26], and the USA recording 120.9 [27]. The data suggest that the mild course of this disease may be typical for the majority of the general population. Hence, the respondents who experienced milder symptoms are more representative for the experience of COVID-19 in the whole population.

The geographical range of the new pandemic, its infectious nature and outcome, meant that individual illness, even in a mild form, generated a number of problems and dilemmas of a non-medical nature due to social and organizational influences [20,21,24,28]. The way COVID-19 is experienced is explained by the category of the patient’s role, understood as a multifaceted construct relating to the struggle with the illness’ symptoms, therapy, and contact with medical institutions. During our analysis, we also identified broader determinants of coping with COVID-19, defined as *ontological uncertainty* and *the coupling of the global and individual dimensions*.

As far as *ontological uncertainty* is considered, it is important to emphasize that, although every disease is a crisis situation for an individual, with regard to COVID-19, the functioning in the role of a patient was largely determined by “the effect of novelty” of this disease. It further deepened and broadened the “uncertainty effect” accompanying the illness. In the case of COVID-19, the margin of uncertainty was particularly wide, ranging from the lack of reliable information on the origin of the disease, risk factors, viral proliferation pathways, and its course, to the lack of effective treatments. This uncertainty was exacerbated by the deficits that became apparent in the healthcare system concerning the procedures for dealing with positive patients. There was also the awareness of numerous unknowns and unpredictable non-medical consequences of COVID-19. All these elements created a climate of instability and even confusion. *Ontological uncertainty*, inspired by the A. Gidden’s concept of “ontological security” [29], signifies functioning in conditions of disturbed stability of the basic foundations of undertaken actions, which under “normal” circumstances are perceived as an axiom. The contagious nature of COVID-19 causes fear of transmission as well; to colleagues, neighbors, and above all, relatives. From this perspective, the role of the patient expands and stretches beyond individual experience, thus accumulating various issues and dilemmas. The picture of struggle with the disease presented by the respondents can even be described as *“being sick in the disease-infected environment”*.

Described experiences were characteristic of the pandemic’s beginnings and additional, time-delayed studies are needed to see if its further course will transform the diagram of the role of the patient that we proposed. However, the results of up-to date research, focused, among others, on the identification of the origin of the new coronavirus, the paths of its spread and evolution, and the search for effective methods of COVID-19 treatment [30,31], do not guarantee a sense of security and stabilization, in the near or distant future. On the contrary, the unpredictability of the epidemiological situation brings about a high level of uncertainty that affects all aspects of existence, becoming, in extreme cases, the cause of panic or suicide attempts [32,33]. As the narratives of respondents reveal, a second negative test result and release from isolation did not always result in a clear self-assessment of their health condition. It did not exclude the fear of infecting someone or of re-infection, which in the light of the then research was not confirmed [34,35,36]. In social relations, as the respondents claimed, suspicion and distance appeared towards the sick and also the convalescents, something also noted by other researchers [6,7,37,38]. Interestingly, the respondents themselves were sometimes not sure whether they actually experienced negative reactions from their environment or just interpreted them as such because they were aware of how much people were afraid of being infected in the initial period of the pandemic [39]. At that time, the spiral of uncertainty and anxiety was fueled by mass media bombarding the recipient with information about the statistics of COVID-19 infections and deaths, the severe course of the disease, and its serious consequences. The COVID-19 pandemic has, hence, started to turn into an “infodemy”, with vast amounts of fake news, deepening disinformation, and social panic [40,41,42]. As we noted from our own research, such media reports, combined with the deficit of scientific knowledge about the disease, intensified the feeling of anxiety concerning one’s own health and the health of loved ones, as well as disorientation in the role of the patient, ultimately placing the respondents in the ontological uncertainty category. This condition was further aggravated by the fact that medical institutions, whose task is to control phenomena related to health and disease and to normalize the behavior of individuals, failed to fulfil their functions effectively due to the lack of uniform standards of conduct in the case of COVID-19. This state of affairs may indicate the depth of the general systemic disorientation in the face of the first wave of the pandemic, a phenomenon that was characteristic not only for Poland [43,44]. Lack of adequate mechanisms of response of medical institutions to the challenges of the pandemic exposed the failure of local systems, while pointing to the weakness of global public health institutions, as emphasized by other authors [45,46].

Such systemic deficits, when analyzed at the macro-social level, have their repercussions on the micro-social scale in determining individual courses of illness. This was particularly noticeable by the respondents, and we categorized it as the *coupling of the global and individual dimensions.* Its significant manifestation was also the “institutionalization” of the process of being ill and subjecting it to certain formal rigors. The unconditional necessity of social isolation, although necessary from the point of view of the public good, was one of the most difficult experiences of being ill for our respondents and their families. As indicated by many authors, social isolation during the pandemic may be a significant risk factor for mental health disorders and the occurrence of post-traumatic stress disorder (PTSD) [47,48,49].

It seems particularly interesting that, despite numerous inconveniences, uncertainties, institutional barriers, and manifestations of social exclusion, these people decided to become plasma donors. This altruistic motivation found in our research is also one of the most important, confirmed, motivational factors in studies conducted in different countries [50]. It appears that developing recruitment campaigns should be focused more on the gratitude and reciprocity that donors feel, along with a focus on an opportunity of helping others.

The value of our research is that, in the narratives of 10 respondents, we performed an in-depth analysis of subjective experiences resulting from the role of a COVID-19 patient during the first stage of the pandemic. As most of the publications related to this disease and SARS-CoV-2 virus to date focused on the severe course of the disease, we characterized its mild course (which is most likely typical for the majority of those infected) in patients until they reached a convalescent status and made the decision to donate plasma. The COVID-19 epidemiological situation is an example of a social process consisting of successive stages, therefore it was essential to capture its earliest stage, which in turn may be helpful in establishing a social response to emergencies model.

This type of work may become a starting point for monitoring the behavior of patients and/or convalescents at subsequent stages of an epidemiological threat, e.g., in a situation of confirming the seasonal nature of infections. The scope of the presented research may also be broadened by the selection of respondents from various social and demographic groups, or those recovering from a severe course of the disease—more so as the grounded theory applied for the purpose of the present work has an elastic and modifiable nature. Such methodological assumptions are very important in relation to phenomena that are dynamic, because even the emergence of new data will allow them to be harmoniously combined with results gathered to date. The highlighted categories can also become the basis for further quantitative research regarding COVID-19 and other new contagious diseases.

Due to the development of research on plasma derived therapy [51,52], our study results are also important in understanding the motivation behind willingness to donate blood for therapeutic or scientific purposes.

The adopted methodology, although creating opportunities for the continuation and extension of the scope of research, simultaneously imposed some limitations. Some of them are related to the methodology of qualitative research. For instance: the ever-deeper exploration of data meant that the analysis grew to include new, more detailed categories. This entailed the temptation to continue the ‘theorizing stage’, but also the risk of ‘never surfacing’ from the data. The difficulty was to definitely stop the analysis and look at the acquired results from a distance. It was, therefore, an effort to control the entire research process so as to obtain the appropriate saturation of all the categories, and on the other hand, not to treat the data too superficially. However, entering the data has led to the emergence of a multitude of closely related threads that we have put together under the common label of the role of the patient as a stable taxonomic framework that organized the presented material.

It was impossible to refer to each of the dimensions of the role of the patient that we described in the study. Hence, the present discussion was conducted at a certain level of generality, referring only to the most important categories identified during our research.

Limitations of our study also resulted from difficulties associated with the specificity of the first wave of the pandemic (e.g., imposed restrictions, the procedure of becoming a convalescent, i.e., the requirement of two negative test results), difficulties with direct contact (necessity of telephone interviews), and a small number of people who went through COVID-19 (who reached the first convalescence status and who decided to donate plasma).

## 5. Conclusions

The disease, despite its mild course, generated a number of non-medical issues, and the entire process of being ill was burdened with emotional and institutional struggles.The role of a COVID-19 patient during the first stage of the pandemic has been defined by: *ontological uncertainty, the coupling of the global and individual dimensions*, and *being sick in the disease-infected environment*. The experience of mild COVID-19 is significantly modified by disease institutionalization. The decisions of sanitary authorities and the results of genetic tests have a strong influence on the subjective feelings of the beginning and end of the disease.Results may contribute to a better understanding of the psychosocial dimension of COVID-19, in particular, the specificity of functioning in it and recovery from it, and to the development of a strategy to support patients during subsequent waves of the COVID-19 pandemic and convalescents’ motivations for plasma donation.

## Figures and Tables

**Figure 1 ijerph-19-06147-f001:**
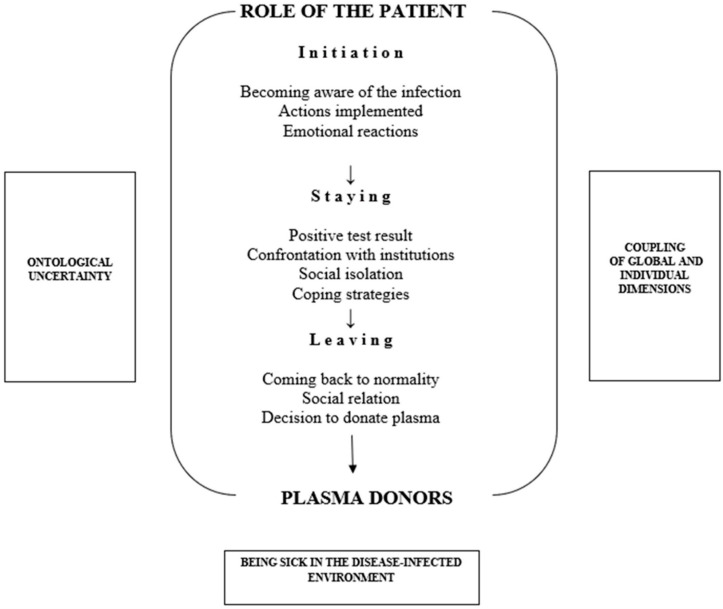
Map of experiences if COVID-19 convalescents.

## Data Availability

The data generated during and/or analyzed during the current study are available from the corresponding author on reasonable request.

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
