# Peer review of "Convalescents’ Reports on COVID-19 Experience—A Qualitative Study"

_ijerph, 2022, doi:10.3390/ijerph19106147_

Round 1

Reviewer 1 Report

The manuscript is clear and relevant for the field of research. The results provide an advance in current knowledge, because the most of the publications on this topic focused on the sever course of the diseases. Therefore, the present paper is valuable to be published, yet, the introduction should be more related to similar research (if it is the case, similar methodology) and could be improved by adding previous research and results on Covid patients. 

Author Response

We would like to thank the reviewer for careful reading of our manuscript and the positive feedback.

The manuscript is clear and relevant for the field of research. The results provide an advance in current knowledge, because the most of the publications on this topic focused on the sever course of the diseases. Therefore, the present paper is valuable to be published, yet, the introduction should be more related to similar research (if it is the case, similar methodology) and could be improved by adding previous research and results on Covid patients.

We have done our best to apply all suggestions and comments into manuscript. We improved the introduction part by adding a few similar studies on COVID-19 patient’s experiences [line 44-58]. Additionally, we have also enriched the literature in the discussion section.

Reviewer 2 Report

I would appreciate seeing described in more detail how did you reach saturation of the sample.

It is somewhat mentioned in Discussion/Limitations but in the Data Analysis (93-95) there is just one unclear sentence "After reaching the saturation... we decided to include all 10 reports". 

Author Response

I would appreciate seeing described in more detail how did you reach saturation of the sample.

It is somewhat mentioned in Discussion/Limitations but in the Data Analysis (93-95) there is just one unclear sentence "After reaching the saturation... we decided to include all 10 reports".

We would like to thank the reviewer for careful and thorough reading of our manuscript and for suggestions, which helped to improve the quality of our manuscript.

We have corrected several linguistic errors and improved English style (corrections are in red).

We have added a short explanation referring to saturation rules in the method’s section [120-126]

Reviewer 3 Report

I read with great interest the paper entitled "First Convalescents' Reports on COVID-19 Experience". A few months after Barney Galland Glaser passed away, it was really fortunate for me to read a good paper based on the research methodology he founded. I would recommend publication of the study after the authors consider my observations below.

- Title: I would ask the authors to add to the title that this is a quality research.

- Introduction: I would ask the authors to give more details about Grounded Theory and the constructivist variation of GT. It is possible that the reader is unfamiliar with the terms and the theory.

-Discussion: At the end of the discussion the authors include limitations of the study. I would ask the authors to separate in the text, the limitations related to the theoretical direction of the article from the limitations of this study.

Author Response

I read with great interest the paper entitled "First Convalescents' Reports on COVID-19 Experience". A few months after Barney Galland Glaser passed away, it was really fortunate for me to read a good paper based on the research methodology he founded. I would recommend publication of the study after the authors consider my observations below.

We would like to thank the reviewer for careful and thorough reading of our manuscript and for the thoughtful comments and constructive suggestions, which helped to improve the quality of our manuscript. Additionally, we appreciate the positive feedback from the reviewer.

We have done our best to include all suggestions and comments in the manuscript.

- Title: I would ask the authors to add to the title that this is a quality research.

Thank you for the suggestion. We have added this information into the title.

- Introduction: I would ask the authors to give more details about Grounded Theory and the constructivist variation of GT. It is possible that the reader is unfamiliar with the terms and the theory.

Thank you for this comment. We have added a few pieces of information regarding Grounded Theory (Introduction section)

-Discussion: At the end of the discussion the authors include limitations of the study. I would ask the authors to separate in the text, the limitations related to the theoretical direction of the article from the limitations of this study.

We have added more information about the limitations in the study and separated it from the limitations related to the theoretical direction of the article (the last part of discussion section).

Reviewer 4 Report

I read the article by Boguszet al. with great interest. Covid is still a hot topic. 

Comments:
In the introduction, the authors focus on the study design. What is missing are the concepts that will be investigated in the study. In fact, it would be important for the authors to make the reader understand why it is important to talk about this topic after two years since the first wave.

When writing materials and methods please refer to the COREQ checklist.

In materials and methods, the description of the sample (page 2 from line 62 to line 78) should be included in the first part of the results. 

In sample, there should be the inclusion and exclusion criteria, the type of sampling and how the methodological rigor was determined.

Author Response

We would like to thank the reviewer for careful and thorough reading of our manuscript, constructive suggestions and the positive feedback. We did our best to include all suggestions and comments in the manuscript.

Iread the article by Boguszet al. with great interest. Covid is still a hot topic.

Comments:

In the introduction, the authors focus on the study design. What is missing are the concepts that will be investigated in the study. In fact, it would be important for the authors to make the reader understand why it is important to talk about this topic after two years since the first wave.

Thank you very much for the comment. We have added more explanations in the introduction section, enriched literature, and improved conclusions. As we have written in the manuscript, our findings may become useful in monitoring the behaviour of patients and/or convalescents at subsequent stages of the epidemiological threat, in other contagious diseases and in enrolment of donors for plasma therapy.

When writing materials and methods please refer to the COREQ checklist.

Mostly, the important information referring to study design, e.g. theoretical framework, participant selection, description of sample and data collection  have already been included in the “Material and Method” section. Additionally we have uploaded the COREQ checklist during resubmission. 

In materials and methods, the description of the sample (page 2 from line 62 to line 78) should be included in the first part of the results.

When preparing the manuscript for IJERPH, we assumed that the sample description (as the material of study) should be located in the “Material and Methods” section and the Results section should only be devoted to the obtained results.

In sample, there should be the inclusion and exclusion criteria, the type of sampling and how the methodological rigor was determined.

We have added more detailed explanation regarding the inclusion and exclusion criteria in the Method section.

Round 2

Reviewer 4 Report

I would like to thank the authors for responding to the suggestions given. The paper is more fluent and understandable for the reader. I do not require further revisions of the paper.